# Cigarette Smoke and Morphine Promote Treg Plasticity to Th17 via Enhancing Trained Immunity

**DOI:** 10.3390/cells11182810

**Published:** 2022-09-08

**Authors:** Ying Shao, Fatma Saaoud, William Cornwell, Keman Xu, Aaron Kirchhoff, Yifan Lu, Xiaohua Jiang, Hong Wang, Thomas J. Rogers, Xiaofeng Yang

**Affiliations:** 1Department of Cardiovascular Sciences, Lewis Katz School of Medicine, Temple University, Philadelphia, PA 19140, USA; 2Center for Inflammation and Lung Research, Lewis Katz School of Medicine, Temple University, Philadelphia, PA 19140, USA; 3Center for Metabolic Disease Research, Lewis Katz School of Medicine, Temple University, Philadelphia, PA 19140, USA

**Keywords:** cigarette smoke, morphine, cigarette smoke plus morphine, regulatory T cells (Treg), knowledge-based transcriptomic analysis

## Abstract

CD4^+^ regulatory T cells (Tregs) respond to environmental cues to permit or suppress inflammation, and atherosclerosis weakens Treg suppression and promotes plasticity. However, the effects of smoking plus morphine (SM + M) on Treg plasticity remain unknown. To determine whether SM + M promotes Treg plasticity to T helper 17 (Th17) cells, we analyzed the RNA sequencing data from SM, M, and SM + M treated Tregs and performed knowledge-based and IPA analysis. We demonstrated that (1) SM + M, M, and SM upregulated the transcripts of cytokines, chemokines, and clusters of differentiation (CDs) and modulated the transcripts of kinases and phosphatases in Tregs; (2) SM + M, M, and SM upregulated the transcripts of immunometabolism genes, trained immunity genes, and histone modification enzymes; (3) SM + M increased the transcripts of Th17 transcription factor (TF) RORC and Tfh factor CXCR5 in Tregs; M increased the transcripts of T helper cell 1 (Th1) TF RUNX3 and Th1-Th9 receptor CXCR3; and SM inhibited Treg TGIF1 transcript; (4) six genes upregulated in SM + M Tregs were matched with the top-ranked Th17 pathogenic genes; and 57, 39 genes upregulated in SM + M Tregs were matched with groups II and group III Th17 pathogenic genes, respectively; (5) SM + M upregulated the transcripts of 70 IPA-TFs, 11 iTregs-specific TFs, and 4 iTregs-Th17 shared TFs; and (6) SM + M, M, and SM downregulated Treg suppression TF Rel (c-Rel); and 35 SM + M downregulated genes were overlapped with Rel^−/−^ Treg downregulated genes. These results provide novel insights on the roles of SM + M in reprogramming Treg transcriptomes and Treg plasticity to Th17 cells and novel targets for future therapeutic interventions involving immunosuppression in atherosclerotic cardiovascular diseases, autoimmune diseases, transplantation, and cancers.

## 1. Introduction

Cigarette smoke is a major cause of death from cancers, cardiovascular disease, and pulmonary disease [1]. CD4^+^FOXP3^+^ regulatory T cells (Treg) have been shown to be increased in the pulmonary lymphocyte follicles of chronic obstructive pulmonary disease (COPD) patients [2]. In addition, exposure to cigarette smoke in the early COPD development leads to a reduction in the signal transducer and activator of transcription 5 (STAT5)^+^, phospho-STAT5 (pSTAT5)^+^ cells, and expression levels of anti-inflammatory/immunosuppressive cytokines such as transforming growth factor- β (TGF-β) and interleukin-10 (IL-10), followed by an increase in STAT3^+^ and pSTAT3^+^ cells and upregulated IL-17 cytokine [3].

Naïve CD4^+^ T cells can be differentiated/polarized into several terminally differentiated T helper cell (Th) subsets including Th1, Th2, Th3, Th5, Th9, follicular T (Tfh), Tfh-13, Th17, Treg, Th22, Th25, CD4^+^ cytotoxic T cells (CD4^+^ CTL), tissue resident memory T cells (Trm), circulating effector memory T cells (Tem), central memory T cells (Tcm), CD28^null^ T cells as well as other T cell subsets [4,5,6,7,8,9,10,11]. Tregs represent a subset of CD4^+^ T cells that express the Forkhead Box P3 (FoxP3) transcription factor (TF) and the high-affinity interleukin-2 (IL-2) receptor (CD25). Tregs are immunosuppressive and control pathogenic effector T cells (Teffs) in autoimmune diseases [12]. A new report showed that three maternal Tregs are classified as CD25^high^Foxp3^+^, PD1^high^IL-10^+^, and TIGIT^+^Foxp3^dim^ [13]. In addition, a single cell RNA Sequencing (scRNA-Seq) report showed that splenic Treg can be classified into six clusters including S100a4^high^S100a6^high^ cluster 1 (activated), Itgb1^high^ cluster 2 (activated), Dusp2^high^Nr4a1^high^Foxp3^high^IL2ra^high^ cluster 3 (activated), Ikzf2^high^Foxp3^high^ cluster 4 (resting), Bach2^high^ cluster 5 (resting), and Satb1^high^Sell^high^ cluster 6 (resting) [14,15]. Moreover, another scRNA-Seq report in characterizing Treg from lymph nodes and non-lymphoid tissue such as skin and colon showed that seven subsets of Tregs can be identified including Tcf7^+^Bcl2^+^Sell^+^Ccr7^+^S1pr1^+^ central Treg (cTreg), Tnfsf4^+^Tnfsf9^+^Cd83^+^Pdcd1^+^Ikzf2^+^ effector Treg (eTreg), Ctla4^+^Rora^low^Itgae^+^ Treg non-lymphoid tissue (NLT)-like, Ctla4^+^Rora^+^Ccr7^high^S1pr1^high^ Treg (lymphoid tissue (LT)-like, Dgat2^+^Lgals3^+^Itgae^+^ skin-specific Treg NLT, Itga4^+^Gimap6^+^ colon-specific Treg NLT, and Il10^+^Gzmb^+^Ccr2^+^ Treg suppressive [16]. In the analysis of Tregs using mass flow and 26 markers, a total of 22 human Treg subsets can be identified [17]. These reports suggest that interactions between antigen epitopes-dependent T cell receptor signaling [18] and independent innate immune stimuli signaling pathways play critical roles in driving naïve Th0 polarization/differentiation into Treg and other Th subsets [19].

Treg plasticity is referred to the capacity of Tregs to acquire the functional characteristics of effector T cells such as T-helper (Th)1, Th2, Th17, or follicular helper T cells while maintaining Foxp3 expression which is known as Th-like Treg [20]. Treg plasticity in atherosclerosis has been reported in aortas of a Western diet (WD)-fed apolipoprotein E (ApoE)^−/−^ mice [21,22] and prolonged exposure to inflammatory cytokines such as interferon-γ (IFNγ), IL 12, and IL-27 via direct activation of the phosphoinositide 3-kinases (PI3K)-protein kinase B (AKT) and Forkhead box protein O1 (Foxo1)/3 pathway [23]. Other reports showed that prolonged exposure to a hyperlipidemic environment promotes the conversion of ApoB antigen-specific Tregs to atherogenic Th17 and Th1-like cells with inflammatory cytokine secretion [24].

Immune responses via innate immune macrophages [25], antigen-specific responses [26,27,28,29,30,31,32,33,34,35,36,37], CD4^+^Foxp3^+^ Treg [7,19,38,39,40], and co-signaling and immune checkpoint receptors [41] play significant roles in modulating inflammation, autoimmunity, and tumor growth. Cigarette smoke (SM) affects both innate and adaptive immunity, playing dual functions in regulating immune responses and leading to dysfunctional innate and adaptive host immunity such as T cell impairment, Treg reduction, dysregulation of the inflammatory response, and causing several inflammatory diseases including respiratory diseases, COPD, and atherosclerotic cardiovascular diseases [42,43,44,45,46].

Cigarette smoking often accompanies illicit drug use, and cigarettes may serve as a drug cue and relapse trigger. Morphine has been shown to impair the innate immune response, T cell activation, and shift toward CD4^+^ Th2 differentiation with increased CD4^+^ Th1 cell death as well as upregulation of different inflammatory chemokines and their receptors [47,48]. It has been reported that morphine inhibits NF-κB signaling in activated T cells of addicts and enhances activated T cell apoptosis, and the effects of morphine T cell suppression are accompanied by elevation of IL-10 and reduction of IL-17 secretion from cultured CD4^+^ T cells [49,50,51]. In addition, cigarette smoke and opioids induce immune cell activation and combination of both can further promote immune system activation. Recently, we reported that smoke plus morphine reduced numbers of Treg cell in the lymph node and lung. We also showed that smoke plus morphine re-shaped Treg cell transcriptome and induced activation of a TNF-like weak inducer of apoptosis (TWEAK), PI3K/AKT, and oxidative phosphorylation (OXPHOS) pathways and a shift to Th17 immunity [52]. However, whether cigarette smoke and morphine modulate Treg plasticity to Th17 and the underlying mechanisms remain poorly understood.

In this study, we analyzed the RNA-sequencing data from cigarette smoke (SM), morphine (M), and cigarette smoke plus morphine (SM + M) treated Tregs with a knowledge based analysis (KBA) plus ingenuity pathway analysis (IPA), which included two parts: (I) phenotypic analysis including 1176 cytokines and their interactors, chemokines (KBA), IPA annotated kinases and phosphatases (KBA), 373 clusters of differentiation (CDs), cell–cell interaction signaling pathways (KBA), 61 regulators of seven CD4^+^ T cell subsets including Th1, Th2, Th9, Th17, Tregs, Tfh, and Th22 (KBA); (II) molecular mechanisms including 266 immunometabolism genes (KBA), 101 trained immunity genes (KBA), and 164 histone modification enzymes (epigenetic regulators, KBA), IPA annotated transcription factors (TFs) in Treg and inducible Treg TFs, and loss of function approach to determine 304 TF expressions in 31 transcriptomic datasets (including the ones from 17 gene knockouts) (Figure 1). We reported that SM + M, M, and SM treated Tregs upregulated the transcripts of cytokines and their interactors, chemokines, kinases, CDs, immunometabolism genes, trained immunity genes, and histone modification enzyme genes. We also showed that SM + M modulated the transcript expressions of the regulators of seven CD4^+^ Th subsets and increased the transcripts of Th17 TF RORC and Tfh factor CXCR5 in Tregs; however, M increased the transcripts of Th1 TF RUNX3 and Th1/Th9 receptor CXCR3, and SM inhibited Treg TGIF1 transcript. In addition, SM + M upregulated genes were matched with Th17 pathogenic genes. SM + M upregulated the transcripts of IPA annotated TFs, iTregs specific TFs, and iTregs Th17 TFs and some of the SM + M upregulated TFs were not upregulated in M and SM alone. Furthermore, SM + M, M, and SM downregulated the transcript expression of Treg suppression TF Rel (c-Rel) and SM + M downregulated genes were overlapped with Rel deficient Tregs downregulated genes. Finally, SM + M specifically reshaped Tregs transcriptomes were not only significantly different from that of SM specific, M specific, Treg versus (vs.) T effectors (Teff), Treg vs. naïve, but also different from that of the Tregs from 28 Treg microarray datasets. Our results provide novel insights on the roles of cigarette smoke plus morphine in reprogramming Treg transcriptomes and Treg plasticity to Th17 cells as well as novel targets for the future therapeutic interventions involving Treg immunosuppression in atherosclerotic cardiovascular diseases, inflammations, autoimmune diseases, transplantations, and cancers.

## 2. Materials and Methods

### 2.1. Statistical Analysis of RNA-Seq Data

Data analysis was carried out using the statistical computing environment R, the Bioconductor suite of packages for R, and RStudio [53]. Raw data were background subtracted, variance stabilized, and normalized by robust spline normalization. Differentially expressed genes were identified by linear modeling and Bayesian statistics using the Limma package [54]. For comparisons between two groups, two-tailed Student’s *t* test was used for evaluation of statistical significance. All original RNA seq data were deposited in the NCBI’s Gene Expression Omnibus database (GSE198210).

### 2.2. Ingenuity Pathway Analysis

We utilized ingenuity pathway analysis (IPA, Qiagen, (https://digitalinsights.qiagen.com/products-overview/discovery-insights-portfolio/analysis-and-visualization/qiagen-ipa/ accessed on 15 June 2021) to characterize clinical relevance and molecular and cellular functions related to the identified genes in our microarray analysis. Differentially expressed genes were identified and uploaded into IPA for analysis. The core and pathways analysis were used to identify molecular and cellular pathways, as we have previously reported [55,56,57]. A *p*-value < 0.05 and a |Z-score| ≥ 1 were set as cutoffs in this study. Of note, pathways with |Z-score| ≥ 2 were designated as significantly influenced.

### 2.3. Metascape Analysis

Metascape (https://metascape.org/gp/index.html#/main/step1, accessed on 20 June 2021) was used for determining signaling pathways involved. This website contains the core of most existing gene annotation portals.

## 3. Results

### 3.1. Cigarette Smoke Plus Morphine, Morphine, and Cigarette Smoke Upregulate the Transcripts of 44, 21, and 8 Cytokines and Their Interactors in Tregs, Respectively, to Promote Plastic/Dysfunctional Tregs

We hypothesized that Treg dysfunctions induced by cigarette smoke (SM), morphine (M), and cigarette smoke plus morphine (SM + M) result in generating specific cytokines, interactors/signaling pathways, and chemokines (Figure 2A). To test this hypothesis, we analyzed the expression changes of cytokine transcripts in SM, M, and SM + M Tregs after screening for the transcripts of total 1176 cytokines and their interactors and 200 chemokines collected from a comprehensive protein database (https://www.proteinatlas.org/search/cytokine accessed on 10 May 2019), as we reported [15]. As shown in Figure 2B, SM + M specifically upregulated the transcripts of 44 cytokines and their interactors including Treg suppressive function weakening AKT1 [58], Th17 transcription factor retinoic acid-binding receptor gamma (RORC), and proinflammatory cytokine tumor necrosis factor (TNF, a trained immunity readout) [59]. Of note, pleiotropic cytokine TNFα proves an inflammatory cytokine or immunosuppressive one by acting on TNF receptor 1 (TNFR1) or Treg promoting TNFR2 [60]. Furthermore, our results showed that SM + M reduced the expression of IFNγ, which limited type 1 CD4^+^ T helper cell (Th1) differentiation and potentially favored the Th17 polarization. However, we did not find any significant changes in the expression of other IL-17 cytokine family members due to their barely detectable levels. SM specifically upregulated the transcripts of eight cytokines and their interactors including Treg suppression weakening phosphoinositide-3-kinase regulatory subunit 1 (PIK3R1) [58]. M specifically upregulated the transcripts of 21 cytokines and their interactors including Treg promoting C-C motif chemokine receptor 5 (CCR5) [40], Th1 cytokine IFNγ, Th2-inducing cytokine IL4, and activated Treg cluster 2 marker integrin subunit beta 1 (ITGB1) [14,19].

In addition, SM + M overlapped with SM in upregulating the transcripts of 19 cytokines and their interactors including MHC class-I component beta-2-microglobulin (B2M), Treg suppression weakening glycolysis gene hexokinase 1 (HK1) [59], inflammasome activating signal 2 purinergic receptor purinergic receptor P2X 7 (P2RX7) [61]. CKLF-like MARVEL transmembrane domain containing 6 (CMTM6) is a key regulator and stabilizer [62] of programmed cell death-1 (PD-1), ligand (PD-L1), and an important immune checkpoint inhibitor [63]. IL-16 is one of the nuclear danger-associated molecular patterns (DAMPs) or nuclear alarmins, which include high-mobility group box-1 protein (HMGB1), IL-33, IL-1*α*, IL1-F7b, and IL-16 and can also bind to pattern recognition receptors (PRRs) and cause a harmful aseptic inflammatory response [64]. Lymphotoxin beta (LTB) promotes chronic inflammation and tertiary lymphoid organ neogenesis [65]. SM overlap with M in upregulating two genes including CD4 and secreted phosphoprotein 1 (SPP1). SPP1, also known as osteopontin (OPN), is a key regulator of tumor progression and immunomodulation involved in multiple pathologies such as cardiovascular, diabetes, kidney, proinflammatory, fibrosis, nephrolithiasis, wound healing, and cancer [66,67]. SM + M overlapped with M in upregulating three genes including protein kinase CAMP-activated catalytic subunit beta (PRKACB), autoimmune regulator proteinase-activated receptor 2 (PAR2, F2RL1) [68], and microtubule interacting and trafficking domain containing 1 (MITD1). All three groups of Tregs shared the upregulation of NF-kB-inhibiting autoimmune regulator [68] Cul3-Kelch like family member 21 (KLHL21) E3 ubiquitin ligase [69]. Moreover, SM + M specifically upregulated the transcripts of three chemokines including FXYD domain containing ion transport regulator 5 (FXYD5), inflammasome adaptor PYD and CARD domain containing (PYCARD) [70], and galectin 9 (LGALS9). M specifically upregulated the transcripts of three chemokines including C-X-C motif chemokine receptor 6 (CXCR6), peptidyl arginine deiminase 2 (PADI2), and C-X-C motif chemokine receptor 3 (CXCR3). SM + M overlapped with M in upregulating integrin subunit beta 3 (ITGB3) (Figure 2C).

To examine the signaling pathways, we used the metascape pathway analysis and showed that SM + M, M, and SM upregulated the transcripts of cytokines in splenic CD4^+^ Tregs have 20, 13, and 3 signaling pathways, respectively. In addition, the SM + M upregulated cytokine group in Tregs has 10 signaling pathways overlapped with that of SM in Tregs (Figure 2D).

Taken together, these results have demonstrated that: (i) SM + M upregulate the transcripts of 44 cytokines and their interactors to promote Tregs toward plastic Th17-Tregs; (ii) SM upregulates the transcripts of 8 cytokines and their interactors, SM overlapped with SM + M upregulates the transcripts of 19 cytokines and their interactors to promote inflammatory/dysfunctional Tregs; (iii) M upregulates the transcripts of 21 cytokines and their interactors, SM overlapped with M upregulate the transcripts of 2 cytokines and their interactors, SM + M overlapped with M upregulate the transcripts of 3 cytokines and their interactors; and (iv) all three groups of Tregs upregulate the transcripts of autoimmune regulator KLHL21.

### 3.2. Cigarette Smoke Plus Morphine, Morphine, and Cigarette Smoke Upregulate the Transcripts of 12, 2, and 10 and Downregulate the Transcripts 18, 16, and 14 Kinases in Splenic Tregs, Respectively; Also, Upregulate the Transcripts of 4, 2, and 6 and Downregulate the Transcripts of 6, 7, and 1 Phosphatases in Splenic Tregs, Respectively

It has been reported that cigarette smoke increases phosphorylation of mitogen-activated protein kinase (MAPK) [71]. We hypothesize that Treg dysfunctions induced by SM, M, and SM + M result in modulation of the transcripts of protein kinases and phosphatases. The gene annotation from IPA showed that SM + M, M, and SM upregulated the transcripts of 12, 2, and 10 kinases and downregulated the transcripts of 18, 16, and 14 kinases in splenic Tregs, respectively (Figure 3A). The metascape pathway analysis for the SM + M upregulated kinases showed the top three upregulated pathways including nucleoside diphosphate metabolic process, interconversion of nucleotide di-and triphosphate, and protein phosphorylation (Figure 3B). Furthermore, our results showed that SM + M, M, and SM upregulated the transcripts of 4, 2, and 6 phosphatases and downregulated the transcripts of 6, 7, and 1 phosphatases in splenic Tregs, respectively (Figure 3C). The metascape pathway analysis of the SM + M upregulated phosphatases showed only one pathway protein phosphorylation (Figure 3D). These data indicate that SM + M upregulate the transcripts of kinases more than that of phosphatases to increase cytokine production and induce Treg dysfunction (Figure 3E).

### 3.3. Cigarette Smoke Plus Morphine, Morphine, and Cigarette Smoke Upregulate the Transcripts of 14, 17, and 1 Clusters of Differentiation (CDs) in Tregs, Respectively, to Promote Weakened, Plastic/Dysfunctional Tregs

Tregs express specific clusters of differentiation (CDs) such as CD4, CD25, CD127, which help easier identification of Tregs from other cell types [39,72,73,74,75,76] and can be defined by specific monoclonal antibodies [77]. Some Treg signature genes that we reported are also CDs [15,40,65]. Since CDs are membrane proteins that mediate outward cell–cell interactions and inward intracellular signaling pathways [41], we hypothesized that SM, M, and SM + M modulate the expressions of CDs in Tregs. We screened the expression changes of 373 CDs identified [15] in three groups of Tregs. As shown in Figure 4A,B, SM + M specifically upregulate 14 CDs in Tregs including immune checkpoints CD96 and PDCD1 (PD-1) [41], Treg suppression-promoting CD52 [78], and resting cluster 6 Treg marker SELL [14]. SM specifically upregulate the transcript of one CD, which is plastic Treg-promoting MUC1 [79]. M specifically upregulates the transcripts of 17 CDs including 3 chemokine receptors, 3 integrins, and TNFRSF14, a TNF receptor superfamily member. Using the metascape pathway analysis we found that the 14 SM + M upregulated CDs have 7 signaling pathways, however, the 17 M upregulated CDs have 9 signaling pathways.

Taken together, these data have demonstrated that (i) SM + M, M, and SM upregulate the transcripts of 14, 17, and one CD, respectively; (ii) SM + M specifically upregulate the transcripts of two immune checkpoints, CD96 and PDCD1 (PD-1); and (iii) SM + M upregulated CDs have 7 signaling pathways and M upregulated CDs have 9 signaling pathways.

### 3.4. Cigarette Smoke Plus Morphine, Morphine, and Cigarette Smoke Upregulate the Transcripts of 21, 1, and 6 Immunometabolism Genes; 5, 4, and 2 Trained Immunity Genes, Respectively; Also, SM + M, M, and SM Upregulate the Transcripts of 9, 1, and 1 and Downregulate the Transcripts of 11, 4, and 3 Histone Modification Enzymes in Splenic Tregs, Respectively

Innate immune cells can develop exacerbated long-term immune responses and an inflammatory phenotype after brief exposure to a primary stimulus, which results in an enhanced and hyperactive inflammatory response towards a second challenge after the return to a nonactivated state. This phenomenon is known as trained immunity or innate immune memory [59,80]. Trained immunity can occur in several traditional innate immune cells such as monocytes/macrophages, natural killer (NK) cells, dendritic cells (DCs), aortic cells, innate immune functions of T cells, and Treg cells, as well as in non-traditional immune cells such as vascular smooth muscle cells (VSMCs), endothelial cells (ECs), hepatocytes, and fibroblasts [59,81,82,83,84,85]. Trained immunity can be induced by several pathogen-associated molecular patterns (PAMPs)/danger-associated molecular patterns (DAMPs) stimuli such as lipopolysaccharides (LPS), β-glucan, Bacillus Calmette–Guerin (BCG), oxidized low-density lipoprotein (ox-LDL), and high-fat diet [86,87,88,89]. Post-translational histone modifications are a key epigenetic mark characterizing trained immunity [90]. It has been reported that cigarette smoke affects both innate and adaptive immune responses and induces metabolic reprogramming and epigenetic chromatin modification [44,91,92]. We hypothesized that SM, M and SM + M modulate trained immunity and histone modification in Tregs. We identified a 266 immunometabolism gene list from previous publications and a 101 trained immunity related gene list from the most updated trained immunity database (http://www.ieom-tm.com/tidb/browse accessed on 10 June 2021). Our data showed that SM + M, M, and SM Tregs significantly upregulated the transcripts of 21, 1, and 6 out of 266 immunometabolism genes and downregulated the transcripts of 6, 4, and 5 out of 266 immunometabolism genes, respectively (Figure 5A,C). The metascape pathway analysis of the SM + M upregulated 21 immunometabolism genes showed a top 5 pathway including purine ribonucleotide metabolic process, ATP metabolic process, pyruvate metabolic process, acyl-CoA metabolic process, and carbon metabolism (Figure 5B). Most of these pathways are considered as key pathways involved in trained immunity [59]. We also found that SM + M, M, and SM significantly upregulated the transcripts of 5, 4, and 2 out of 101 trained immunity genes (Figure 5E). Since TNF-α, AKT, and mevalonate kinase (MVK) are the hallmark genes in trained immunity [93], SM + M upregulates trained immunity pathways more than SM and M acting alone.

Our previous report indicated that tissue Tregs upregulate as much as 80% of signaling pathways, being innate immune pathways in response to tissue microenvironments [15]; that innate immune memory (trained immunity) is established via histone modifications including histone 3 lysine 14 (H3K14) acetylation [59,80,94,95,96]; and that metabolic diseases downregulate the majority of 164 histone modification enzymes and upregulate a few enzymes [97]; therefore, histone posttranslational modifications are a key feature of trained immunity [91] (Figure 5F). We hypothesized that SM, M and SM + M stabilize their effects on Treg transcriptomes by modulating the expression of transcripts of histone modification enzymes. We screened the expression changes of the transcripts of 164 histone modification enzymes in three groups of Treg transcriptomes, as we reported [97]. The 164 histone modification enzymes were classified into seven groups: 55 histone methyltransferases, 24 histone demethylases, 31 histone acetyltransferases, 18 histone deacetylases, 15 histone serine kinases, 8 histone ubiquitination enzymes, and 13 histone SUMOylation enzymes [97]. As shown in Figure 5G, SM upregulated the transcripts of histone methyltransferase SET and MYND domain containing 1 (SMYD1); M upregulated the transcripts of histone deacetylase histone deacetylase 11 (HDAC11); and SM + M upregulated the transcripts of two histone acetyltransferases, TATA-Box binding protein associated factor 5 Like (TAF5L) and N-alpha-acetyltransferase 10 (NAA10); two deacetylases, sirtuin 7 (SIRT7) and histone deacetylase 5 (HDAC5), two methyltransferases, Ehmt2 (EHMT2) and PR/SET domain 16 (PRDM16), two histone serine kinase protein phosphatases, 1 catalytic subunit gamma (PPP1CC) and VRK serine/threonine kinase 1 (VRK1), and one SUMOylation enzyme protein inhibitor of activated STAT 3 (PIAS3).

Taken together, these results have demonstrated that cigarette smoke plus morphine, morphine, and cigarette smoke upregulate a few histone modification enzyme genes in five types of enzymes including acetyltransferases, deacetylases, methyltransferases, serine kinases, and SUMOylation enzymes, but not demethylases and ubiquitination enzymes. The upregulated histone modification enzyme genes may collaborate with upregulated transcription factors to promote transcription of genes in Tregs.

### 3.5. Cigarette Smoke Plus Morphine Increase the Transcripts of Th17 Transcription Factor (TF) RORC and CXCR5; Morphine Decreases Treg TF FOXP3, and IKZF4 and Increases Th1 TF RUNX3; and SM Inhibits Th1, Th17 and Treg Linage Marker TGIF1 in Tregs; and Six out of 139 SM + M Upregulated Genes Were Matched with Th17 Pathogenic Genes and 57, 38 out of 139 SM + M Upregulated Genes Were Matched with Group II and Group III Th17 Pathogenic Genes, Respectively

Our previous studies reported the screening of 61 regulators of seven CD4^+^ T helper cell subsets including Th1, Th2, Th9, Th17, Tregs, Tfh, and Th22 in tissues and various gene deficient transcriptomic datasets of Tregs, and found that while HDAC6 and Bcl6 are important regulators of Treg plasticity, GATA3 determines the fate of plastic Tregs by controlling whether it will convert in to either Th1-Tregs or antigen-presenting cell (APC)-Tregs [19]. Th17 cells produce IL-17A, IL-17F, IL-21 and IL-22 and play a critical role in driving autoimmune tissue inflammation [92]. Th17 cells are differentiated by a combination of cytokines including IL-1, IL-6, IL-23, and TGFβ1 to induce transcription factor ROR required for the generation of Th17 cells [98]. RAR related orphan receptor C (RORC) is the first lineage defining transcription factor for Th17 [98,99,100,101].We hypothesized that SM + M, M, and SM modulate the transcript of CD4^+^ Th subset regulators in Tregs and promote Treg plasticity toward Th17. As shown in Figure 6A, SM + M increased the transcripts of Th17 transcription factor RORC and Tfh linage factor CXCR5 and inhibited the transcripts of resting Treg cluster 5 marker BTB domain and CNC homolog 2 (BACH2), Th9-Th17-Treg linage factor TGFB induced factor homeobox 1 (TGIF1), CD28, and cytotoxic T-lymphocyte associated protein 4 (CTLA4) in Tregs. M upregulated the transcripts of Th1 transcription factor RUNX family transcription factor 3 (RUNX3) and Th1-Th9 C-X-C motif chemokine receptor 3 (CXCR3) and downregulated the transcripts of Treg TF *FOXP3*, Th2-Th9-Tfh TF interferon regulatory factor 4 (IRF4), Treg TF IKAROS family zinc finger 4 (IKZF4) [15], Treg signaling molecule signal transducer and activator of transcription 5A (STAT5A), Th9-Th17-Treg lineage factor TGIF1, Th9 C-C motif chemokine receptor 3 (CCR3), Th17 receptor CCR6, CCR8 and CTLA4. However, SM downregulated the transcript of Th9-Th17-Treg lineage factor TGIF1, which was also downregulated by M and SM + M. Of note, we also examined the expression changes of transcription factors of Th1/Treg, Th2/Treg, Tfh/Treg in SM + M group. We did not find significant changes in the transcriptomic expression of TBX21, GATA3 or IRF4, BCL6 TFs for Th1/Treg, Th2/Treg, Tfh/Treg, respectively. Taken together, these results have demonstrated that SM + M promote Treg plasticity toward Th17; M promotes Treg plasticity toward Th1; SM does not significantly promote Treg plasticity but weakens Tregs (Figure 6B).

To further confirm that SM + M promote Treg plasticity towards Th17, we collected the top 100 ranked Th17 pathogenic genes from single cell RNA sequencing data of other publications [102], then we examined the overlapped genes with our 139 upregulated genes from SM + M Tregs. We found that six genes including TNF, SAM, and SH3 domain containing 3 (SASH3), IL-16, MAPK activated protein kinase 3 (MAPKAPK3), programmed cell death protein 1 (PDCD1), and inhibitor of DNA binding 3 (ID3) were matched with Th17 pathogenic genes (Figure 6C). Furthermore, we detected the matched genes between 139 upregulated genes from SM + M Tregs and the upregulated genes from TGFβ3 + IL6 induced Th17 and IL1β + IL6 induced Th17 (group II pathogenic Th17) as well as TGFβ3 + IL6 + IL23 induced Th17 and IL1β + IL6 + IL23 induced Th17 (group III pathogenic Th17) [103]. Our results showed that 57 genes out of 139 upregulated genes in SM + M treated Tregs were matched with group II Th17 pathogenic genes, and 38 genes out of 139 upregulated in SM + M treated Tregs were matched with group III Th17 pathogenic genes (Figure 6D). These data further confirmed that SM + M promote Treg plasticity toward Th17.

### 3.6. Cigarette Smoke Plus Morphine, Morphine, and Cigarette Smoke Upregulate the Transcripts of 70, 16, and 24 and Downregulate the Transcripts of 75, 39, and 12 IPA-Annotated Transcription Factors (TFs) in Tregs, Respectively; and SM + M, M, and SM Modulate the Transcripts of 22, 17, and 14 out of 149 Inducible Treg (iTreg) Specific TFs and 14, 8, and 5 out of 78 iTregs and Th17 Shared TFs in Mouse Splenic Tregs, Respectively

To further test our hypothesis, we then pulled out all TFs, those devoted to major roles in determining the molecular mechanisms in the reprogramming of Treg transcriptomes. We used the ingenuity pathway analysis (IPA) database and identified 304 TFs modulated in the three groups of Treg transcriptomes (*p*-value < 0.05). We found that SM + M upregulated the transcripts of 98 TFs, among which 70 TFs (71.4%) were SM + M specific. M upregulated the transcripts of 27 TFs, among which 16 TFs (59.3%) were M specific. SM upregulated the transcripts of 53 TFs, among which 24 TFs (45.3) were SM specific. Furthermore, SM + M downregulated the transcripts of 108 TFs, among which 75 TFs (69.4%) were SM + M specific. M downregulated the transcripts of 68 TFs, among which 39 TFs (57.4%) were M specific. SM downregulated the transcripts of 38 TFs, among which 12 TFs (31.6%) were SM specific (Figure 7A,B). The metascape pathway analysis showed that SM + M specific upregulated TFs have 20 signaling pathways, M specific upregulated TFs have 3 signaling pathways, and SM specific upregulated TFs have 16 signaling pathways (Figure 7C–E). Moreover, to directly examine the effects of SM + M, M, and SM in Treg specific transcription machinery, we examined the expression changes of 149 inducible Tregs (iTregs) specific TFs and 78 iTregs-Th17 shared TFs that others previously reported [104]. As shown in Figure 7F,G, SM + M upregulated the transcripts of 11 out of 149 iTreg TFs including histone deacetylase 5 (HDAC5), YY1 associated factor 2 (YAF2), MAX dimerization protein 4 (MXD4), PBX homeobox interacting protein 1 (PBXIP1). In addition, SM + M upregulated the transcripts of 4 out of 78 iTregs-Th17 shared TFs including PYD and CARD domain containing (PYCARD), PTTG1 regulator of sister chromatid separation, securin (PTTG1), PDZ, and LIM domain 1 (PDLIM1), which were not upregulated in SM-Tregs and M-Tregs (red arrows indicated in Figure 7G). SM upregulated one iTreg TF activating transcription factor 5 (ATF5) which is SM specific. M specifically downregulated the transcripts of seven iTreg TFs including cysteine and serine rich nuclear protein 1 (CSRNP1), CCAAT enhancer binding protein beta (CEBPB), CAMP responsive element modulator (CREM), MAX dimerization protein 1 (MXD1), FOXP3, IKAROS family zinc finger 4 (IKZF4), TSC22 domain family member 3 (TSC22D3) which are M specific. SM downregulated the transcripts of two iTregs-Th17 favored FOSL2 and ID2. The metascape pathway analysis of SM + M upregulated 15 iTreg-specific TFs identified in the top seven pathways (Figure 7H). The Venn diagram analysis indicated that SM + M Tregs overlaps one transcription factor (TF) pathway with that of SM Tregs TFs and M Tregs TFs; SM + M Tregs TFs overlap three TF pathways with that of SM Tregs TFs, and SM Tregs TFs overlaps one TF pathway with that of iTregs TFs (Figure 7I).

In addition, our data showed that SM + M, M, and SM downregulated the transcripts of Treg suppression TF Rel (c-Rel) (Figure 7J). To find out the functional significance of transcription factor c-Rel downregulation in SM + M, M, and SM Tregs, we reasoned that if c-Rel is critical in promoting Treg generation and suppressive function, Rel deficiency in Tregs will lead to downregulation of significant numbers of genes in Tregs. Through our search in NIH-NCBI GeoDatasets database, we found that an Australian group published a paper on RNA Seq datasets from c-Rel deficiency (KO) in Tregs (GSE154166) [105]. We found that 35 genes were shared by the downregulated gene list from SM + M Tregs and the downregulated gene list from c-Rel knockout (KO) Tregs (Figure 7K). The results suggest that the expressions of 35 genes are promoted by c-Rel function in Tregs. Since c-Rel is an essential transcription factor for Treg immunosuppressive function, downregulation of c-Rel in SM + M Tregs is another key finding on Treg weakening induced by SM + M treatment.

In addition, we generated a biclustering heat map to determine the expression changes of 304 transcription factors annotated by ingenuity pathway analysis (Figure 8A) in SM + M, M, SM, and other 28 Treg gene mutant transcriptomic datasets (loss of function approach to identify causative roles of Treg master regulators underlying Treg transcriptomic reprogramming) including cluster 1 (activated Tregs) transcription factor Blimp1 (PRDM1) deficiency (BLIMP1-knockout, KO) Tregs (GSE27143) [14,106], histone acetyltransferase p300 (cluster 1 activated Treg regulator) KO Tregs (GSE47989) [107], resting cluster Tregs regulator BATCH2-KO Tregs (GSE45975) [108], activated Tregs suppressing FOXO1-KO Tregs (GSE40657) [109,110], T helper 17 (Th17) versus (vs.) naïve, Treg vs. Teff, Treg vs. T naïve, FOXP3^+^ follicular T helper (FOXP3 Tfh) vs. Teff, follicular regulatory T cells (Trf) vs. Treg [111], iTreg (72 h (hr)), Th17 (72 hr) (GEO: GSE90570) [104], microRNA maturation enzyme dicer deficiency (Treg Dicer-KO) (GSE11818) [112], CD69-CD62L^low^ (activated T cell markers) vs. CD62^high^ (resting T cell marker) (GSE36527) [113], visceral adipose tissue Tregs transcription factor peroxisome proliferator activated receptor gamma (PPARγ) mutant vs. PPARγ wild-type (WT) (GSE37532) [114], Treg pro-survival cytokine IL2-KO (GSE14350) [115], Th2 transcription factor GATA binding protein 3 (GATA3)-KO Tregs (GSE39864) [116], Treg IKAROS family zinc finger 4 (IKZF4, EOS)-KO, GATA1-KO Tregs, X-box binding protein 1 (XBP1)-KO Tregs (GSE40273) [117], histone deacetylase 9 (HDAC9)-KO Treg (GSE36095) [118], IL10-KO Treg, Epstein–Barr virus-induced 3 (EBI3, IL35b, IL-27b)-KO Tregs, EBI3/IL10-DKO (GSE29262) [119], iTreg vs. failed iTreg, activated Treg vs. native Treg, native Treg vs. FOXP3-KO, and activated Treg vs. FOXP3-KO (GSE14415) [120], T cell immunoreceptor with Ig and ITIM domain (TIGIT) positive vs. negative Treg (GSE56299) [121]. As shown in Figure 8B,C, the transcript expressions of SM + M modulated 304 TF genes in Tregs were regulated in 28 transcriptomic datasets (cutoff: l FC l > 1). Of note, the 16 datasets with significant modulation of the transcript expressions of more than 10 transcription factors were highlighted in red including BLIMP-KO, BATCH-KO, FOXO1-KO, Th17, Treg, iTreg 72 h, Th17 72 h, Treg Dicer-KO, CD69-CD62^Low^, Treg IL2-KO, IL10-KO, EBI3-KO, EBI3-IL10-DKO, activated Treg, native Treg, and TIGIT^+^ vs. TIGIT^—^. These results have demonstrated that SM + M specifically reshaped Tregs transcriptomes were not only significantly different from that of SM treatment (SM specific), M treatment (M specific), Treg versus T effectors (Teff), Treg vs. naïve, but also different from that of the Tregs from the other 28 Treg microarray datasets.

Taken together, these results have once again demonstrated, in master gene regulation, that: (i) SM + M upregulate the transcripts of 70 TFs, among which 11 were iTregs-specific TFs, and 4 were iTregs-Th17-favored TFs; (ii) M upregulated the transcripts of 16 TFs, but most of them were neither iTreg nor iTregs-Th17 favored; and (iii) SM upregulated the transcripts of 24 TFs, among which 6 were shared with SM + M but to a less extent. These results suggest that SM + M, M, and SM weaken Treg transcription machinery in different TF pathways. Of note, as shown in Figure 7G, the transcript of FOXP3, Treg-specific TF, was downregulated in M-treated Treg. Since there are 1498 FOXP3 target genes in the TF binding site database (http://tfbsdb.systemsbiology.net/searchtf?searchterm=V_FOXP3_01_M01599 accessed on 20 June 2021) [122], M downregulation of FOXP3 has significant effects on Treg transcriptomes.

Taken together, these results have demonstrated that SM, M and SM + M induce iTreg dysfunction and plasticity [123] more than facilitating iTreg development.

## 4. Discussion

Tregs are an immunosuppressive subgroup of CD4^+^ T cells which are identified by the expression of FOXP3. They can secrete cytokines such as IL-10 and TGFβ. The modulation capacity of Treg holds an important role in several diseases such as cardiovascular disease [40,76,124,125,126,127], inflammation [15,57,80,81,128,129], autoimmune diseases [19], cancers [7,38,130,131,132], and transplantation [133]. The imbalance of Tregs is an important factor in the pathogenesis of several smoke (SM) related diseases including COPD [2]. Furthermore, SM increases a person’s risk of using illicit drugs. Morphine (M) can reduce the effectiveness of several functions of both innate and adaptive immunity, and significantly decreases cellular immunity [134]. Previous findings indicate that there is an unexpected plasticity between T effector cells and Tregs [135] and this plasticity might play a critical role in the control of the immune system, enabling a rapid switch from suppression to active immunity and play a key role in the development of several inflammatory diseases including atherosclerosis.

Treg-Th17 plasticity has been reported in other pathological settings. IL-6 and TGF-β stimulation of FOXP3^+^ Treg can induce Th17 producing cells that are dependent on the expression of transcription factors RORγ and RUNX [128,129,136]. However, whether smoke and morphine promote Treg-Th17 plasticity remains poorly characterized. To address this issue, we analyzed the RNA sequencing data from SM, M, and SM + M treated Tregs and performed a knowledge based and IPA approach with the strategies we pioneered [31,62]. We made the following significant findings: (1) SM + M, M, and SM upregulate the transcripts of cytokines/interactors, chemokines, and clusters of differentiation (CDs) in Tregs; (2) SM + M, M, and SM modulate the transcripts of kinases and phosphatases; (3) SM + M, M, and SM upregulate the transcripts of immunometabolism genes, trained immunity genes, and histone modification enzyme genes; (4) SM + M, M, and SM modulate the transcripts of the regulators of seven CD4^+^ T helper cell (Th) subsets including Th1, Th2, Th9, Th17, Treg, follicular Th (Tfh), and Th22; (5) SM + M increase the transcripts of Th17 TF RORC and Tfh factor CXCR5 in Tregs; M increases the transcripts of Th1 TF RUNX3 and Th1-Th9 receptor CXCR3; and SM inhibits Treg TGIF1 transcript; (6) six out of 139 genes upregulated in SM + M treated Tregs were matched with the top ranked 100 Th17 pathogenic genes; (7) 57 and 39 out of 139 genes upregulated in SM + M treated Tregs were matched with group II and group III Th17 pathogenic genes, respectively; (8) SM + M upregulates the transcripts of 70 IPA annotated TFs, 11 iTregs specific TFs, and 4 iTregs-Th17 TFs; M upregulates the transcripts of 16 IPA annotated TFs, 3 iTregs specific TFs, and downregulates the transcripts of 8 iTregs-Th17 specific TFs; and SM upregulates the transcripts of 24 IPA annotated TFs, 7 iTregs specific TF, and one iTreg-Th17 specific TFs; (9) SM + M significantly upregulates the transcript expression of four iTregs TFs which were not upregulated in SM and M treated Tregs; (10) SM + M upregulate the transcripts of three iTreg-Th17 shared TFs, SM upregulate ATF5 and downregulate the transcripts of seven TFs; and (11) SM + M, M, and SM downregulate the transcript expression of Rel (c-Rel) in Tregs and 35 genes out of 1354 SM + M downregulated genes were overlapped with Rel deficient Tregs downregulated genes.

Our previous publication showed that CD4^+^FOXP3^+^ Treg cells have many active innate immune pathways [15,132], and the immunosuppressive functions of Tregs can be sustained, although Treg cell plasticity in chronic inflammatory atherosclerosis has been reported [19,40]. We also reported that Treg cells and other adaptive immune cells not only respond to antigen stimulation [74] but also respond to the DAMPs/PAMPs stimulation as in other innate immune cells [6,7,134,135,137]. Recently, we reported that while histone deacetylase 6 (HDAC6) and follicular Th cell specific transcription factor B-cell lymphoma 6 (Bcl6) are important regulators of Treg plasticity, Th2 specific transcription factor GATA3 determines the fate of plastic Treg by controlling whether it will convert into either Th1-Treg or antigen-presenting cell (APC) Treg [19]; and Treg from spleen, lymph nodes, intestine, and visceral adipose tissues promote tissue repair by generating secretomes similar to those of stem cells; and sharing TFs aryl hydrocarbon receptor (AHR), ETS-variant transcription factor 5 (ETV5), early growth response 1 (EGR1), and Kruppel-like factor 4 (KLF4) with stem cells, and Treg canonical secretomes and transcriptomes may be regulated by 1176 cytokines, 1706 canonical secretomes, kinome (complete list of human genome-encoded 651 kinases), cell surface receptors such as the complete list of 373 CDs, immunometabolism and trained immunity gene list, and the complete list of 1496 transcription factors.

Under normal conditions, Foxp3^+^ Tregs migrate into inflamed tissues to suppress inflammatory responses to exert immunosuppressive effects and accelerate tissue repair [15,138]. In pre-existing respiratory comorbidities such as COVID-19, which leads to the disruption of the immune system, exacerbated inflammation which is partly due to the decreased expression of Tregs or defects in these cells results in weakening the Tregs effects of inflammatory inhibition, causing an imbalance in Treg/Th17 ratio, and increasing the risk of respiratory failure [137,139,140,141]. Since our data showed that the smoke and morphine combination promote weakened, plastic/dysfunctional Tregs and Treg plasticity toward Th17 cells, smoke plus morphine in combination in pre-existing respiratory co-morbidities such as COVID-19 will exacerbate inflammation and increase the severity of the disease.

As shown in Figure 9, we proposed a novel working model to integrate all the findings. More than 7000 chemicals in cigarette smoke and morphine bind to the membrane and intracellular receptors for PAMPs/DAMPs and opioid receptors in the FOXP3^+^ Tregs and induce intracellular signaling pathways [44] and Tregs transcriptomic reshaping. Tregs can be induced by smoke plus morphine to become proinflammatory cells expressing RORC, which lose suppressive capacity while retaining FOXP3 expression. In the presence of smoke plus morphine, Tregs can acquire a Th17-like phenotype. These cells will increase the transcript expression of Th17 transcription factor RORC, which promotes Tregs conversion (plasticity) to Th17-like cells and increases the transcript expression of cytokines and their interactors, chemokines, CDs, kinases, and phosphatases. Furthermore, smoke plus morphine-treated Tregs will increase the transcript expression of trained immunity genes and histone modification enzyme genes, resulting in a decreased immunosuppression function of Tregs and enhanced immune inflammatory response, leading to increased trained immunity as underlying mechanisms contribute to Tregs plasticity to Th17-like Tregs.

One limitation of the current study is that due to the low-throughput nature of verification techniques in the laboratories, we could not verify every result we identified with the analyses of high-throughput data. We believe that extensive future work is needed to verify all the high-throughput results reported here with relatively low-throughput methods currently in most laboratories. Nevertheless, our findings provide novel insights on the roles of tissue Treg in controlling cardiovascular inflammation, immune responses, and promoting tissue repair and regeneration as well as novel targets for future therapeutic interventions for immunosuppression, cardiovascular diseases, inflammations, autoimmune diseases, transplantation, cancers, and tissue repair.

## Figures and Tables

**Figure 1 cells-11-02810-f001:**
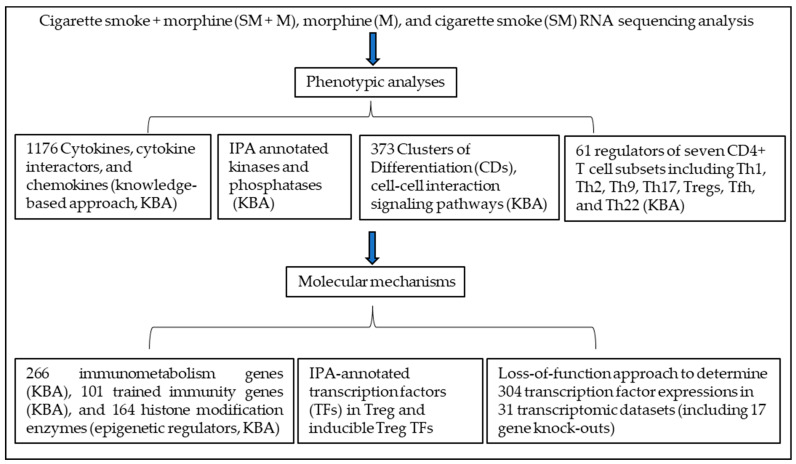
The logic flow of knowledge based analysis (KBA) plus ingenuity pathway analysis (IPA) of Treg transcriptomics modulated in three groups of Tregs: cigarette smoke (SM) Tregs, morphine (M) Tregs, and cigarette smoke plus morphine (SM + M) Tregs includes two parts: (I) phenotypic analysis including 1176 cytokines and their interactors and chemokines (KBA), IPA annotated kinases and phosphatases (KBA), 373 clusters of differentiation (CDs), cell–cell interaction signaling pathways (KBA), 61 regulators of seven CD4^+^ T cell subsets including Th1, Th2, Th9, Th17, Tregs, Tfh, and Th22 (KBA); (II) molecular mechanisms including 266 immunometabolism genes (KBA), 101 trained immunity genes (KBA), and 164 histone modification enzymes (epigenetic regulators, KBA), IPA annotated transcription factors in Treg and inducible Treg TFs, and loss of function approach to determine 304 transcription factor expressions in 31 transcriptomic datasets (including 17 gene knockouts).

**Figure 2 cells-11-02810-f002:**
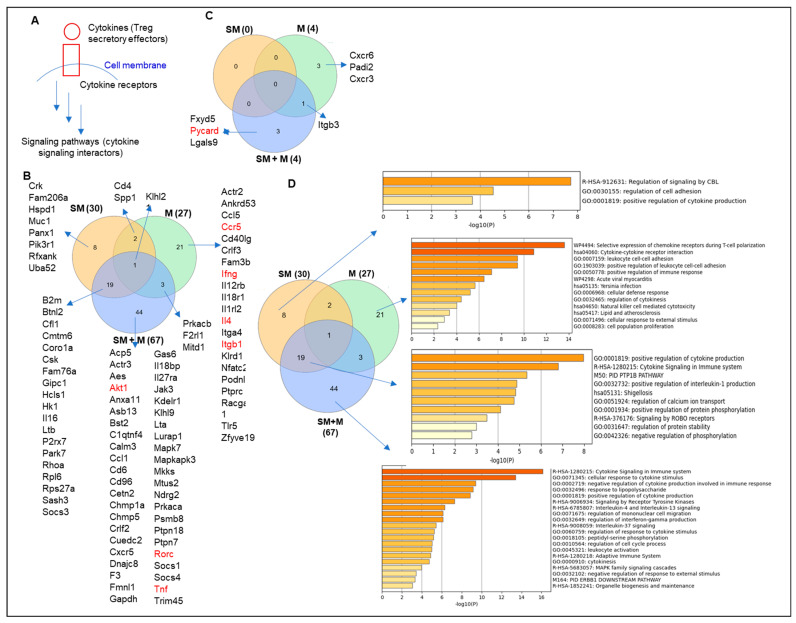
T helper cell 17 (Th17)-specific transcription factor RORC and proinflammatory cytokine TNF are upregulated in Tregs from the cigarette smoke plus morphine (SM + M)-treated group. (**A**) A schematic figure showed the interaction between cytokines and their receptors to induce a signaling pathway. (**B**) The Venn diagram analysis showed that SM + M, morphine (M), and cigarette smoke (SM) specifically upregulated the expressions of 44, 21, and 8 cytokines and their interactors respectively after screening for a total of 1176 cytokines and their interactors (https://www.proteinatlas.org/search/cytokine accessed on 10 May 2019) in splenic CD4^+^ Tregs. (**C**) SM + M, M, and SM specifically upregulated the expressions of three, three, and zero chemokines, respectively, after screening for a total of 200 chemokines in mouse splenic CD4^+^ Tregs. (**D**) Cytokine signaling and innate immune response to LPS are the cytokine pathways upregulated in SM + M treated Tregs. The metascape pathway analysis showed that upregulated cytokines induced by SM + M, M, and SM in splenic CD4^+^ Tregs have 20, 13, and 3 signaling pathways, respectively. In addition, the SM + M upregulated cytokine group in Tregs has 10 signaling pathways overlapped with that of SM in Tregs.

**Figure 3 cells-11-02810-f003:**
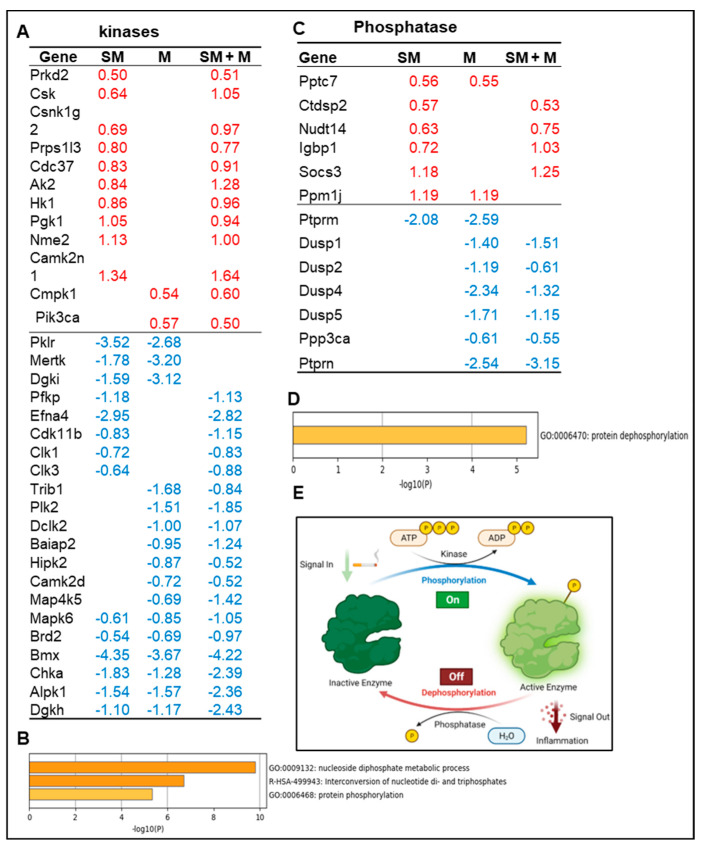
(**A**) The gene annotation from ingenuity pathway analysis (IPA) showed that cigarette smoke plus morphine, morphine, and cigarette smoke upregulated the expressions of 12, 2, and 10 kinases and downregulated 18, 16, and 14 kinases in splenic Tregs, respectively. (**B**) The metascape pathway analysis for the upregulated kinases. (**C**) The gene annotation from IPA showed that cigarette smoke plus morphine, morphine, and cigarette smoke upregulated the expressions of 4, 2, and 6 phosphatases and downregulated 6, 7, and 1 phosphatases in splenic Tregs, respectively. (**D**) The metascape pathway analysis for the upregulated kinases. (**E**) Schematic figure showed the kinases (phosphorylation) and phosphatases (dephosphorylation) of proteins and their effects on inflammation.

**Figure 4 cells-11-02810-f004:**
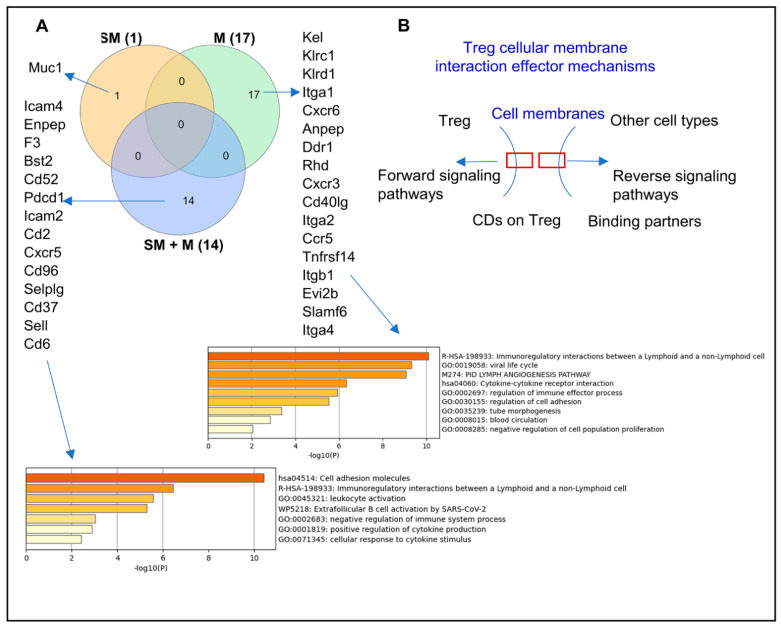
Cell adhesion molecules and leukocyte activation are the CD molecular pathways upregulated in cigarette smoke plus morphine (SM + M) treated Tregs. (**A**) The Venn diagram analysis showed that SM + M, morphine (M), and cigarette smoke (SM) specifically upregulated the expressions of 14, 17, and 1 cluster of differentiation (CD) respectively after screening for a total of 373 CDs in mouse splenic CD4^+^ Tregs. The SM + M upregulated CD group has 7 signaling pathways while the M upregulated CD group has 9 signaling pathways. (**B**) The schematic figure showed the CDs on Treg and the forward signaling pathway, and on the other cell types with a reverse signaling pathway.

**Figure 5 cells-11-02810-f005:**
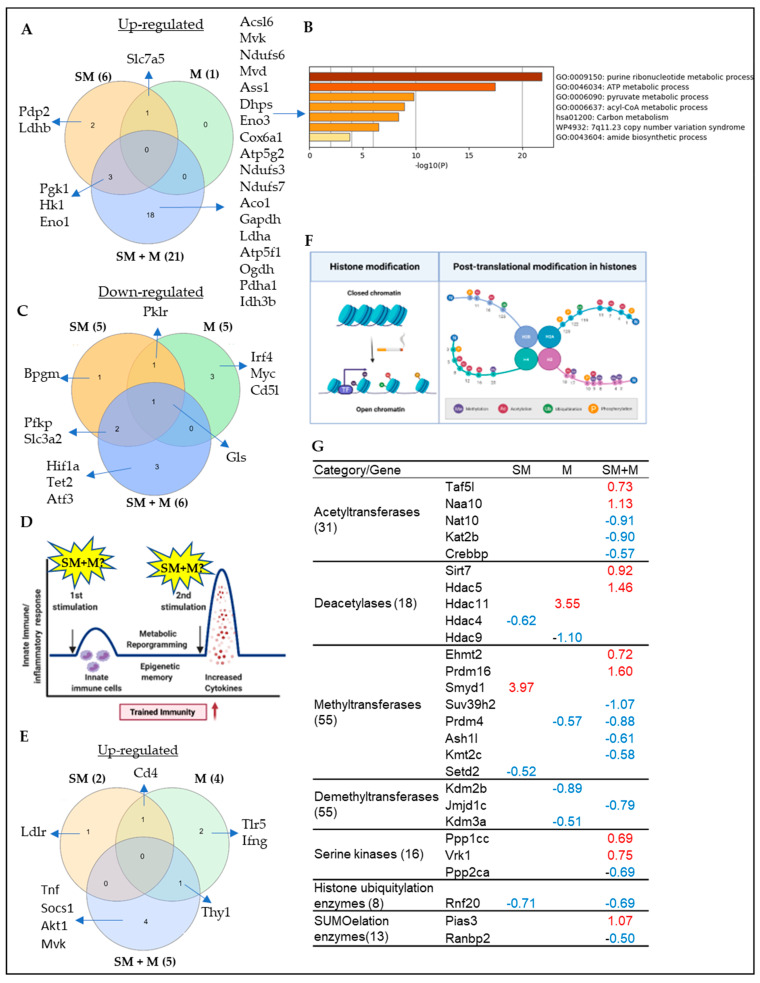
(**A**) The knowledge based analysis indicated that cigarette smoke plus morphine (SM + M), morphine (M), and cigarette smoke (SM) significantly upregulated the transcripts of 21, 1, and 6 genes out of 266 immunometabolism genes. (**B**) The metascape pathway analysis of the SM + M upregulated transcripts of immunometabolism genes. (**C**) SM + M, M, and SM significantly downregulated the transcripts of 6, 5, and 5 out of 266 immunometabolism genes. (**D**) The schematic overview of trained immunity. (**E**) SM + M, M, and SM significantly upregulated the transcripts of 5, 4, and 2 out of 101 trained immunity genes. The 101 trained immunity genes were collected from the new trained immunity database (http://www.ieom-tm.com/tidb/browse accessed on 10 June 2021). (**F**) A schematic figure showed the posttranslational histone modification. (**G**) SM + M, M, and SM upregulated the transcripts of 9, 1, and 1 histone modification enzymes and downregulated the transcripts of 11, 4, and 3 out of 164 histone modification enzymes in splenic Tregs, respectively. The 164 histone modification enzymes were classified into seven groups: 55 histone methyltransferases, 24 histone demethylases, 31 histone acetyltransferases, 18 histone deacetylases, 16 histone serine kinases, 8 histone ubiquitination enzymes, and 13 histone SUMOylation enzymes.

**Figure 6 cells-11-02810-f006:**
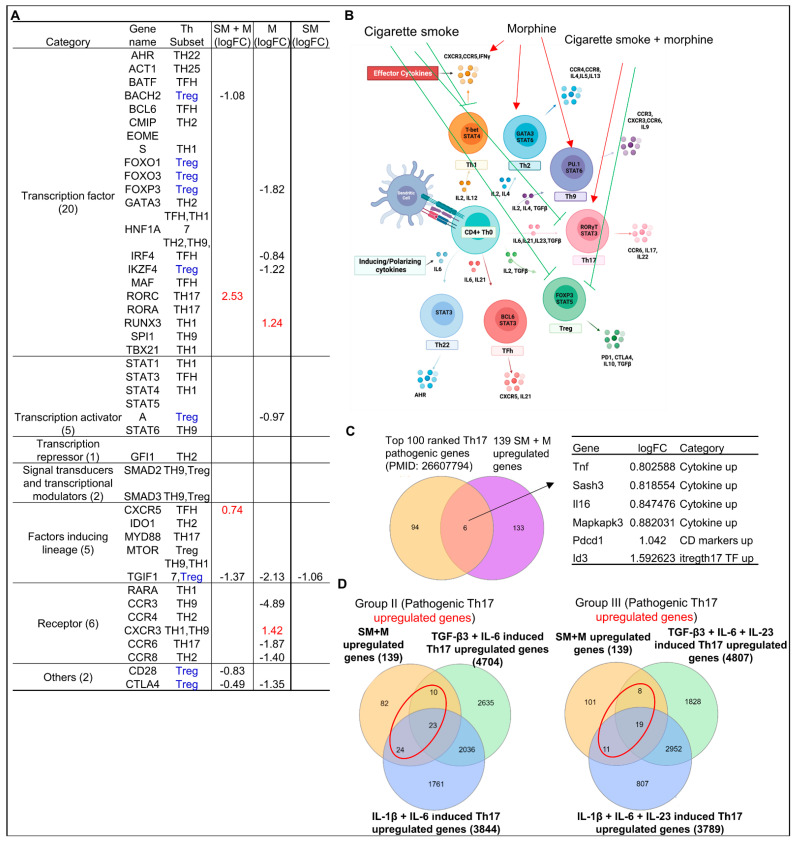
Cigarette smoke plus morphine promoted Th17 genes and suppressed Treg genes. (**A**) SM + M, M, and SM modulated the expressions of 6, 11, and 1 out of 41 regulators of seven CD4^+^ T helper cell (Th) subsets (Th1, Th2, Th9, Th17, Treg, follicular Th (Tfh), and Th22) respectively, which include seven groups such as transcription factors (20), transcription activators (5), transcription repressor (1), signal transducers and transcription modulators (2), factors inducing lineage (5), receptors (6), and others (2). SM + M upregulated Th17 transcription factor RORC and CXCR5 (red); M upregulated transcription factor RUNX3, Th1-Th9 receptor CXCR3; and SM alone did not upregulate any genes but downregulated Treg TGIF1. (**B**) Schematic figure shows CD4^+^ T helper (Th) subset regulators. SM + M promoted Th17 (red arrow) and inhibited Treg (green blockers). M promoted Th1, Th2, and Th9 (red arrow). However, SM inhibited Th1, Th17, and Treg (green blockers). (**C**) The Venn diagram analysis showed that six genes including four cytokines and interaction proteins including TNF, SASH3, IL-16, MAPKAPK3, one CD marker PDCD1, and one Th17-iTreg shared transcription factor ID3 that were upregulated in SM + M treated Tregs were matched with the top-ranked 100 Th17 pathogenic genes previously reported. (**D**) The Venn diagram analysis showed that 57 genes out of 139 upregulated in SM + M treated Tregs (red circle) were matched with the group II Th17 pathogenic genes and 38 genes out of 139 upregulated in SM + M treated Tregs (red circle) were matched with the group III Th17 pathogenic genes previously reported.

**Figure 7 cells-11-02810-f007:**
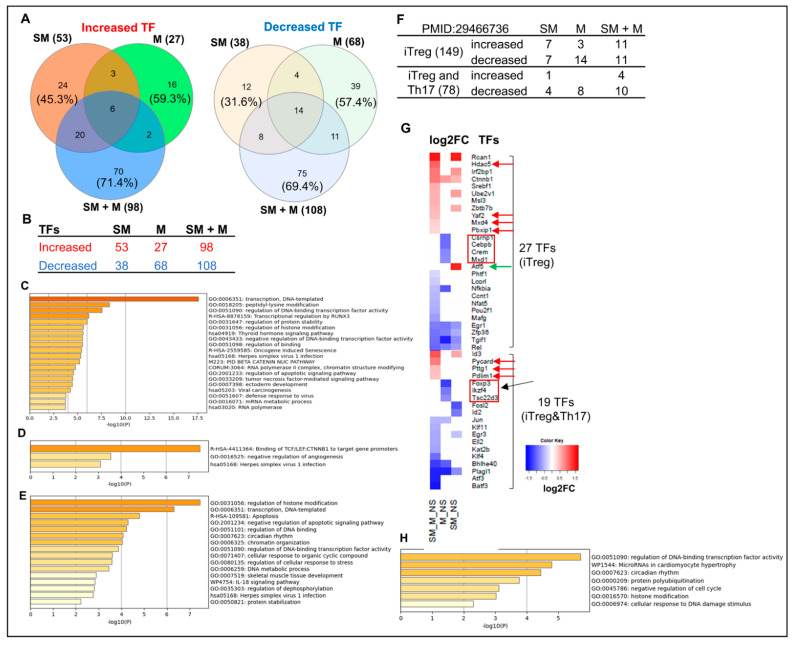
Cigarette smoke plus morphine, morphine, and smoke modulate the expression of transcription factors (TFs). (**A**,**B**) The ingenuity pathway analysis (IPA) showed that SM + M, M, and SM specifically upregulated 70, 16, and 24 and downregulated 75, 39, and 12 IPA-annotated TFs in Tregs, respectively. (**C**–**E**) The metascape pathway analysis of the specifically upregulated genes in SM + M, M, and SM, respectively. (**F**) SM + M, M, and SM modulated the expressions of 22, 17, and 14 out of 149 inducible Treg (iTreg)-specific TFs and 14, 8, and 5 out of 78 iTregs-Th17-shared TFs in mouse splenic Tregs, respectively. (**G**) The heat map was generated by using the value of log2FC of the TFs (*p*-value < 0.05) in each comparison. The results showed that: (1) SM + M significantly upregulates the expression of four inducible Treg (iTregs)TFs including HDAC5, YAF2, MXD4, and PBXIP1, which were not upregulated in SM treated Tregs and M treated Tregs (red arrow). In addition, SM + M upregulated three iTreg and Th17 shared TFs (red arrows) including PYCARD, PTTG1, PDZ, and PDLIM1; (2) SM upregulated TF ATF5 (green arrow) is SM specific; and (3) seven M downregulated TFs including CSRNP1, CEBPB, CREM, MXD1, FOXP3, IKZF4, TSC22D3 are M specific. (**H**) The metascape pathway analysis for SM + M 15 upregulated iTreg-specific TFs. (**I**) The Venn diagram analysis indicated that SM + M Tregs overlaps one TF pathway with that of SM Tregs TFs and M Tregs TFs; SM + M Tregs TFs overlap three TF pathways with that of SM Tregs TFs, and SM Tregs TFs overlaps one TF pathway with that of iTregs TFs. (**J**) SM + M, M, and SM downregulated the expression of Rel (c-Rel) in Tregs (*p* < 0.05). (**K**) 35 genes out of 1354 SM + M downregulated genes were overlapped with Rel-deficient Tregs downregulated gene.

**Figure 8 cells-11-02810-f008:**
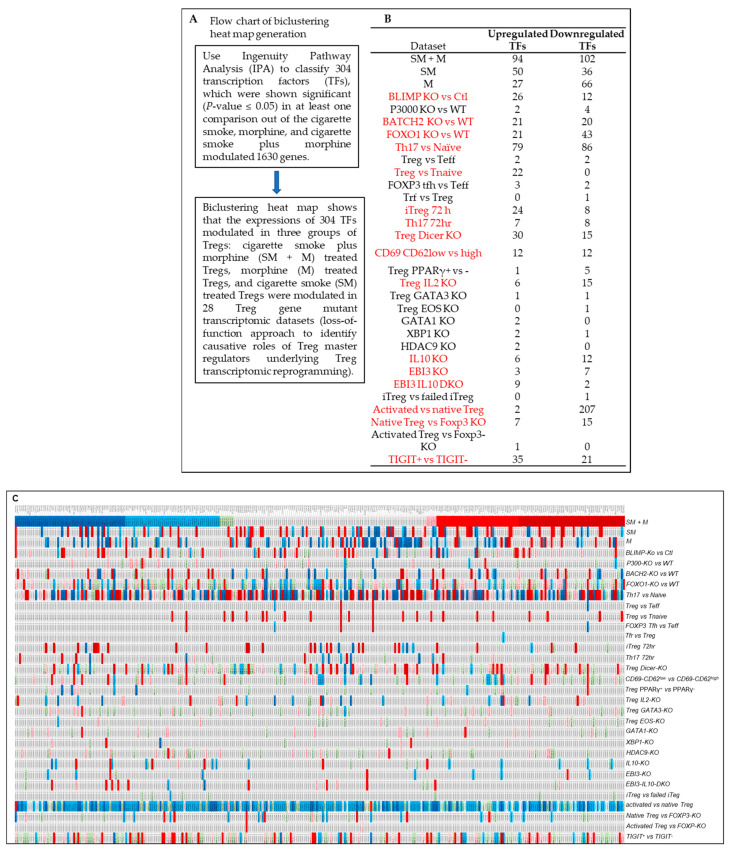
(**A**) Flow chart of biclustering heat map generation. (**B**) The expressions of cigarette smoke plus morphine (SM + M) modulated 304 transcription factor (TF) genes in Tregs were regulated in 28 transcriptomic datasets (cutoff: l FC l > 1). Of note, the 16 datasets with significant modulation of the expressions of more than 10 TFs were highlighted in red including BLIMP-KO, BATCH-KO, FOXO1-KO, Th17, Treg, iTreg 72 h, Th17 72 h, Treg Dicer, CD69-CD62^Low^, Treg IL2-KO, IL10-KO, EBI3-KO, EBI3-IL-10-DKO, activated Treg, native Treg, and TIGIT^+^ vs. TIGIT^−^. (**C**) Biclustering heat map of the expressions of cigarette smoke plus morphine (SM + M) significantly modulated TF genes in Tregs (304 IPA annotated TFs) with *p* < 0.05 from three comparisons). The list of 304 TFs were used to match to other 28 Treg datasets. These results indicated that SM + M specifically reshaped Tregs transcriptomes were not only significantly different from that of smoke treatment (SM specific), morphine treatment (M specific), Treg versus (vs.) T effectors (Teff), Treg vs. naïve, but also different from that of the Tregs from 28 Treg microarray datasets including BLIMP1-KO Tregs (GSE27143), p300-KO Tregs (GSE47989), BATCH2-KO Tregs (GSE45975), FOXO1-KO Tregs (GSE40657), T helper 17 (Th17) vs. naïve, Treg vs. Teff, Treg vs. T naïve, FOXP3 Tfh vs. Teff, Tfr vs. Treg), iTreg (72 h h), Th17 (72 h) (GSE90570), Treg Dicer-KO (GSE11818), CD69-CD62L^low^ (activated T cell markers) vs. CD62^high^ (resting T cell marker) (GSE36527), visceral adipose tissue PPARγ+ vs. PPARγ- (GSE37532), Treg pro-survival cytokine IL2-KO (GSE14350), GATA3-KO Tregs (GSE39864), Treg EOS-KO, GATA1-KO Tregs, XBP1-KO Tregs (GSE40273), HDAC9-KO Treg (GSE36095), IL10-KO Treg, EBI3-KO Tregs, EBI3 and IL10-DKO (GSE29262), iTreg vs. failed iTreg, activated Treg vs. native Treg, native Treg vs. FOXP3-KO, and activated Treg vs. FOXP3-KO (GSE14415), and TIGIT positive vs. negative Treg (GSE56299). The color keys indicate the value of log2 fold changes (FC) of the gene expression (significance); red ranges indicate upregulation of genes, and blue ranges indicate downregulation of genes. Abbreviation: vs.: versus; WT: wild-type; KO: gene-deficient, knockout; DKO: double gene KO; Ctl: controls.

**Figure 9 cells-11-02810-f009:**
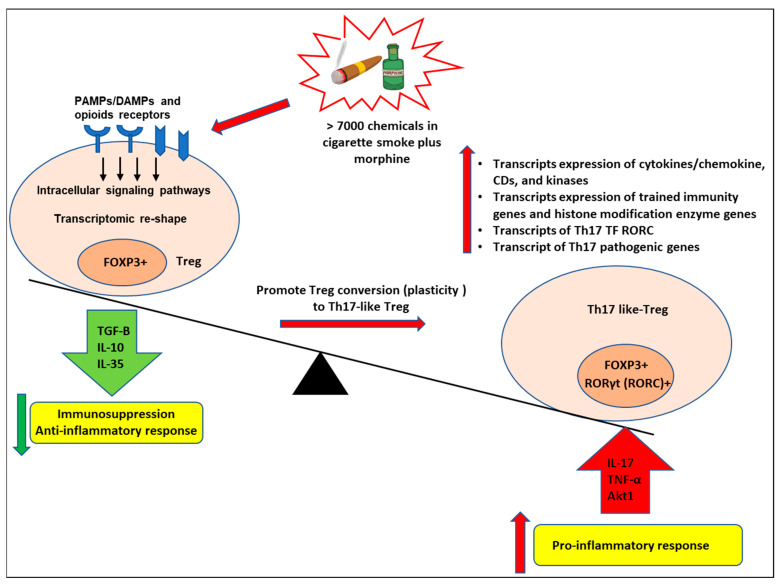
Our working model. More than 7000 chemicals in cigarette smoke and morphine bind to the membrane and intracellular PAMPs/DAMPs and opioid receptors in the FOXP3+ Tregs and induce intracellular signaling pathways and Tregs transcriptomic reshaping. Tregs can be induced by smoke plus morphine to become proinflammatory cells expressing RORC, which lose suppressive capacity while retaining FOXP3 expression. In the presence of smoke plus morphine, Tregs can acquire a Th17-like phenotype. These cells will increase the transcript expression of Th17 transcription factor RORC, which promotes Tregs conversion (plasticity) to Th17-like cells and increases the transcript expression of cytokines and their interactors, chemokines, CDs, kinases, and phosphatases. Furthermore, cigarette smoke plus morphine-treated Tregs will increase the transcript expression of trained immunity genes and histone modification enzyme genes, resulting in a decreased immunosuppression function of Tregs and enhanced immune inflammatory response and leading to increased trained immunity as underlying mechanisms contributing to Tregs plasticity to Th17-like Tregs.

## Data Availability

The RNA sequencing data presented in this study can be found in online repositories. The names of the repository/repositories and accession number(s) can be found below: NCBI, accession ID: GSE198210.

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
