# Peer review of "Cigarette Smoke and Morphine Promote Treg Plasticity to Th17 via Enhancing Trained Immunity"

_cells, 2022, doi:10.3390/cells11182810_

Round 1

Reviewer 1 Report

In the manuscript by Shao et al, the authors investigate the potential CD4+ regulatory T cells (Treg) weakening effects of cigarette smoke and morphine combination via transcriptomic reprogramming. The high-throughput work follows up on previous low tech observations by the other investigators on the damaging effects of cigarette smoke and morphine in T cells and other immune cells. Here, the authors use a broad combination of approaches including Ingenuity pathway analysis and knowledge basis analysis of transcriptome profiling to address their hypotheses, which are numerous. Overall, the work is interesting and the findings are very important. In each of the avenues of investigation there appears to be exciting data that suggest at least some effects of cigarette smoke and morphine combination in the context of Treg transcriptional reprogramming as well as potentially weakened Treg and Treg plasticity toward more proinflammatory CD4+ Th17 subset.

One concern I have is that the authors need to examine a possibility whether cigarette smoke and morphine combination modulate the expressions of Th17 promoting cytokines such as IL-6, IL-23, and IL-21 and inhibitory cytokines including IL-2, IL-27, IL-12 and IFNg (PMID: 23325835).

Author Response

Response to Reviewer 1 comments

Comments and Suggestions for Authors

In the manuscript by Shao et al, the authors investigate the potential CD4+ regulatory T cells (Treg) weakening effects of cigarette smoke and morphine combination via transcriptomic reprogramming. The high-throughput work follows up on previous low tech observations by the other investigators on the damaging effects of cigarette smoke and morphine in T cells and other immune cells. Here, the authors use a broad combination of approaches including Ingenuity pathway analysis and knowledge basis analysis of transcriptome profiling to address their hypotheses, which are numerous. Overall, the work is interesting and the findings are very important. In each of the avenues of investigation there appears to be exciting data that suggest at least some effects of cigarette smoke and morphine combination in the context of Treg transcriptional reprogramming as well as potentially weakened Treg and Treg plasticity toward more proinflammatory CD4+ Th17 subset.

One concern I have is that the authors need to examine a possibility whether cigarette smoke and morphine combination modulate the expressions of Th17 promoting cytokines such as IL-6, IL-23, and IL-21 and inhibitory cytokines including IL-2, IL-27, IL-12 and IFNg (PMID: 23325835).

Thanks a lot for the suggestion. We have examined those cytokines and found that smoke and morphine combination only reduced the expression of interferon-gamma (Ifng), which limited type 1 CD4+ T helper cell (Th1) differentiation and potentially favored the Th17 polarization. However, we did not find any significant changes in the expressions of other IL-17 cytokine family members due to their barely detectable levels.  We have added information to Result 3.1.

Many thanks again for the reviewers’ and editors’ most insightful critiques and advice.

Sincerely,

Xiaofeng Yang, MD, Ph.D., FAHA, Corresponding Author

Reviewer 2 Report

This manuscript by Shao et al reports the RNA-Seq data analysis on Tregs modulated by cigarette smoke and morphine. Although cigarette smoke and morphine modulate immune responses and inflammation through numerous mechanisms, how smoke and morphine reprogram Tregs transcriptomes remains poorly understood. In this aspect, the present findings are very important and relevant. There are many reports on cigarette smoke and morphine induced toxicities of immune systems in low throughput manners, but high-quality high throughput transcriptomic analyses on smoke and morphine effects on Tregs have been lacking. Thus, this manuscript has filled in this significant knowledge gap and is appropriately decorated with relevant and satisfactory figures. I truly enjoyed reading the article. 

Major concerns: 

I would suggest giving some space to the discussions of presented findings in cigarette smoke and morphine modulation on Treg transcriptomes related to pre-existing respiratory comorbidities such as COVID-19 (PMIDs: 35874688; 357860880; 35497875; 35378753; 34936112).

Author Response

Response to Reviewer 2 comments

Comments and Suggestions for Authors

This manuscript by Shao et al reports the RNA-Seq data analysis on Tregs modulated by cigarette smoke and morphine. Although cigarette smoke and morphine modulate immune responses and inflammation through numerous mechanisms, how smoke and morphine reprogram Tregs transcriptomes remains poorly understood. In this aspect, the present findings are very important and relevant. There are many reports on cigarette smoke and morphine induced toxicities of immune systems in low throughput manners, but high-quality high throughput transcriptomic analyses on smoke and morphine effects on Tregs have been lacking. Thus, this manuscript has filled in this significant knowledge gap and is appropriately decorated with relevant and satisfactory figures. I truly enjoyed reading the article. 

Major concerns: 

I would suggest giving some space to the discussions of presented findings in cigarette smoke and morphine modulation on Treg transcriptomes related to pre-existing respiratory comorbidities such as COVID-19 (PMIDs: 35874688; 357860880; 35497875; 35378753; 34936112).

We greatly appreciate the reviewer’s suggestions. We have added the following paragraph to the discussion section “Under normal conditions, Foxp3+ Tregs migrate into inflamed tissues to suppress inflammatory responses to exert immunosuppressive effects and accelerate tissue repair ( PMID: 33613572, PMID: 19770521). In pre-existing respiratory comorbidities such as COVID-19, which leads to the disruption of the immune system, exacerbated inflammation which is partly due to the decreased expression of Tregs or defects in these cells resulting in weakening the Tregs effects of inflammatory inhibition, causing an imbalance in Treg/Th17 ratio, and increasing the risk of respiratory failure (PMID: 32416070, PMID: 34631206, PMID: 35378753, PMID: 35497875). Since our data showed that the smoke and morphine combination promotes weakened, plastic/dysfunctional Tregs and Treg plasticity toward Th17 cells, smoke plus morphine combination in pre-existing respiratory co-morbidities such as COVID-19 will exacerbate inflammation and increase the severity of the disease. 

Many thanks again for the reviewers’ and editors’ most insightful critiques and advice.

Sincerely,

Xiaofeng Yang, MD, Ph.D., FAHA, Corresponding Author

Reviewer 3 Report

This manuscript by Shao et al investigated the roles of cigarette smoke and morphine in functioning as danger associated molecular patterns in re-shaping CD4+ regulatory T cell (Treg) transcriptomes via upregulating the transcripts encoding cytokines, kinases, phosphatases, and transcription factors. Previous reports demonstrated that smoke and morphine damage immune functions. However, how smoke and morphine affect Tregs transcriptomes, the top immunosuppressive cell types in controlling various immune responses and inflammations, remains poorly characterized. In this study, the authors provided convincing results showing that cigarette smoke and morphine promote Treg plasticity to Th17 via enhancing trained immunity. These findings not only revealed new regulatory pathways re-shaping Treg transcriptomes, but also demonstrated a compelling interplay of cigarette smoke and morphine in modulating Tregs and immunosuppressive mechanisms. This study is well designed and performed, and the transcriptomic mechanisms reported here are informative and innovative. I have a few suggestions:

1. Four IL-17 family cytokines were identified (PMID: 28620097). Is it possible to also determine the expression changes of other IL-17 family cytokines in the datasets?

2. The authors may want to examine whether cigarette smoke and morphine affect Treg plasticity to other plastic Tregs such as Th1/Treg, Th2/Treg, Tfh/Treg (PMID: 33322482)?

3. The authors are suggested to check the potential changes in sirtuins family members and ROS regulators in the datasets (PMID: 34950150)?

4. The images in Fig 4A, 6B, 7G look blur in the PDF file. The author may want to increase the resolution of the images.

Author Response

Response to Reviewer 3 comments

Comments and Suggestions for Authors

This manuscript by Shao et al investigated the roles of cigarette smoke and morphine in functioning as danger associated molecular patterns in re-shaping CD4+ regulatory T cell (Treg) transcriptomes via upregulating the transcripts encoding cytokines, kinases, phosphatases, and transcription factors. Previous reports demonstrated that smoke and morphine damage immune functions. However, how smoke and morphine affect Tregs transcriptomes, the top immunosuppressive cell types in controlling various immune responses and inflammations, remains poorly characterized. In this study, the authors provided convincing results showing that cigarette smoke and morphine promote Treg plasticity to Th17 via enhancing trained immunity. These findings not only revealed new regulatory pathways re-shaping Treg transcriptomes, but also demonstrated a compelling interplay of cigarette smoke and morphine in modulating Tregs and immunosuppressive mechanisms. This study is well designed and performed, and the transcriptomic mechanisms reported here are informative and innovative. I have a few suggestions:

  1. Four IL-17 family cytokines were identified (PMID: 28620097). Is it possible to also determine the expression changes of other IL-17 family cytokines in the datasets?

Response 1: We greatly appreciate the reviewers for the insightful suggestion. In our transcriptomic datasets, we did not find significant changes in the expression of the IL-17 cytokine family members due to their barely detectable levels. We have added information to Result 3.1.

  1. The authors may want to examine whether cigarette smoke and morphine affect Treg plasticity to other plastic Tregs such as Th1/Treg, Th2/Treg, Tfh/Treg (PMID: 33322482)?

Response 2: We greatly appreciate the reviewer for raising this issue up. We have examined the expression changes of Transcription factors (TFs) of Th1/Treg, Th2/Treg, Tfh/Treg in SM + M group. We did not find significant changes in the transcriptomic expressions of Tbx21, Gata3 or Irf4, Bcl6 TFs for Th1/Treg, Th2/Treg, Tfh/Treg, respectively. We added this information to Result 3.5.

  1. The authors are suggested to check the potential changes in sirtuins family members and ROS regulators in the datasets (PMID: 34950150)?

Response 3: We thank the reviewer for the suggestion. We have examined the expression changes of the sirtuins family genes and ROS regulators. In our datasets, we only found that the Sirt7 transcript was significantly increased in the smoke and morphine combination group. In addition, we did not find significant changes in the expressions of ROS regulators in SM + M Tregs. However, in the future, we will examine ROS production.    

  1. The images in Fig 4A, 6B, 7G look blur in the PDF file. The author may want to increase the resolution of the images.

Response 4: Thanks a lot for your comments. We have fixed this problem and increased the resolution of the images.

Many thanks again for the reviewers’ and editors’ most insightful critiques and advice.

Sincerely,

Xiaofeng Yang, MD, Ph.D., FAHA, Corresponding Author
